

# Genetic links between endometriosis and cancers in women

Salma Begum Bhyan[1], Li Zhao[2], YongKiat Wee[1], Yining Liu[3] and Min Zhao[1]

[1] School of Science and Engineering, University of the Sunshine Coast, Sunshine Coast, Queensland, Australia
[2] Dongguan Women and Children's Hospital, Dongguan, China
[3] The School of Public Health, Institute for Chemical Carcinogenesis, Guangzhou Medical University, Guangzhou, China

## ABSTRACT

Endometriosis is a chronic disease occurring during the reproductive stage of women. Although there is only limited association between endometriosis and gynecological cancers with regard to clinical features, the molecular basis of the relationship between these diseases is unexplored. We conducted a systematic study by integrating literature-based evidence, gene expression and large-scale cancer genomics data in order to reveal any genetic relationships between endometriosis and cancers in women. We curated 984 endometriosis-related genes from 3270 PubMed articles and then conducted a meta-analysis of the two public gene expression profiles related to endometriosis which identified Differential Expression of Genes (DEGs). Following an overlapping analysis, we identified 39 key endometriosis-related genes common in both literature and DEG analysis. Finally, the functional analysis confirmed that all the 39 genes were associated with the vital processes of tumour formation and cancer progression and that two genes (*PGR* and *ESR1*) were common to four cancers of women. From network analysis, we identified a novel linker gene, *C3AR1*, which had not been implicated previously in endometriosis. The shared genetic mechanisms of endometriosis and cancers in women identified in this study provided possible new avenues of multiple disease management and treatments through early diagnosis.

Corresponding authors
Salma Begum Bhyan,
salma.bhyan@research.usc.edu.au
Min Zhao, mzhao@usc.edu.au

## INTRODUCTION

Endometriosis is a chronic disease of women occurring at the reproductive stage of their lives. Worldwide, approximately 176 million (6–10%) women are affected by this disease (*Kvaskoff, Horne & Missmer, 2017*) diagnosed by the formation of endometrial-like tissues and lesions, not only on the walls of the uterus but also in the fallopian tubes and on the pelvic wall. It is a common disease that can cause pelvic inflammation, chronic pain and infertility (*Giudice & Kao, 2004*). The first report on endometriosis-associated ovarian cancer was published in 1927 (*Sampson, 1927*) and the histological transformation of endometriosis to endometrioid ovarian cancers (EnOC) was reported in 1996 (*De La Cuesta et al., 1996*). This process has been confirmed by subsequent epidemiological studies (*Heidemann et al., 2014*; *Kim et al., 2014*; *Kvaskoff et al., 2015*). The occurrence of synchronous endometriosis in ovarian cancer lesions was reported by *Jimbo et al. (1997)*

with 23.1 and 43% of EnOC. A separate epidemiological study showed a 2–3% higher risk in ovarian cancer patients with endometriosis than controls (*Wei, William & Bulun, 2011*). A significant positive association was observed with different histological subtypes of ovarian cancers, particularly with endometroid, epithelial and clear cell types (*Saavalainen et al., 2018*; *Gandini et al., 2019*; *Pearce et al., 2012*). *Pearce et al. (2012)* reported that endometriosis is associated with an increased risk of clear cell type ovarian cancer. The highest rate of incidence ratio of ovarian cancer was observed among women with ovarian endometriosis, particularly endometrioid and clear cell types (*Saavalainen et al., 2018*). In addition, endometriosis has been found to show more modest association with seromucinous borderline (*Maeda & Shih, 2013*; *Samartzis et al., 2013*), low-grade serous ovarian carcinomas (*Pearce et al., 2012*). Sequencing and immune-histochemical studies have demonstrated a clonal relationship between benign and malignant counterparts suggesting that the cancers may have arisen from the endometriotic lesions (*Anglesio et al., 2015*; *Chene et al., 2015*; *Stamp et al., 2016*), (*Wiegand et al., 2010*).

Endometriosis, along with breast cancer, are estrogen-dependent chronic gynecological disorders and have similar risk factors associated with reproduction and the use of hormone replacement therapy (*Munksgaard & Blaakaer, 2011*). In addition, both endometriosis and cancer exhibit uncontrolled, estrogen-dependent proliferation, invasion, neo-angiogenesis and metastases (*Burney & Giudice, 2012*; *Thomas & Campbell, 2000*). Previous studies have identified the connection between endometriosis and breast cancer (BC) (*Kokcu, 2011*; *Kvaskoff et al., 2015*). In Denmark, data collected from 1977 to 2012 on 114,327 women showed that women who were diagnosed as endometriosis positive before age 40 had a 14% reduced risk of breast cancer than those diagnosed between age 40 and 50. Women diagnosed at age 50 or older were more than twice as likely to develop breast cancer as women of the same age who did not have endometriosis (*Mogensen et al., 2016*). *Bulletti et al. (2010)* reported that 20–25% of patients might be asymptomatic during the diagnosis of endometriosis. Although it is plausible that endometriosis is associated with increased risks for endometrial and breast cancer, epidemiological studies of the association are inconsistent.

Cervical cancer is the abnormal growth of cells in the lining of the cervix. The most common cervical cancer is squamous cell carcinoma (*Marwah et al., 2012*). On the other hand, cervical endometriosis is rare and often shows no symptoms. An epidemiological study of the association between the two diseases has reported the decreased risk of cervical cancer (*Kvaskoff et al., 2015*). A recent study suggested an association between cervical clear cell carcinoma (CCC) and cervical endometriosis (*Hashiguchi et al., 2018*). However, a decreased risk of cervical cancer of squamous cell histology among women with endometriosis has been suggested (*Saavalainen et al., 2018*).

Although endometriosis and endometrial cancer are two separate diseases, epidemiological, biological, and molecular data suggest that there could be links between the two disorders (*Kvaskoff et al., 2015*). Endometriosis is characterized by the formation of endometrial-resembling tissue external to the uterus while endometrial cancer (sometimes called uterine cancer) begins in the layer of cells that form the lining (endometrium) of the uterus. Endometriosis and endometrial cancer are both hormonally regulated
diseases, having a common risk factor (higher levels of estrogen) and similar treatment measures (contraceptive pill and hormonal therapies) (*Wetendorf & Demayo, 2012*). Both are involved in increased risk of uterine fibroids (*Rowlands et al., 2011*; *Uimari, Järvelä & Ryynänen, 2011*). The predominant symptom of endometrial cancer is pelvic pain, which is also one of the main symptoms of endometriosis. In a recent genetic study, it was identified that these two diseases have common genetic causes (*Painter et al., 2018*). However, some previous studies showed the opposite or no significant association. *Kvaskoff et al. (2015)* reported eight studies on the association between endometriosis and endometrial cancer and that there was no association in five studies where numbers of endometrial cancer cases ranging from 12 to 97 (*Brinton et al., 1997*; *Brinton et al., 2005*; *Melin, Sparen & Bergqvist, 2007*; *Venn et al., 1999*). Two studies showed an association between endometriosis and increased endometrial cancer risk (*Melin et al., 2006*; *Zucchetto et al., 2009*) but one study suggested a significant inverse association (*Borgfeldt & Andolf, 2004*).

Therefore, a detailed investigation on the genetic control of the different cancer types along with endometriosis is necessary in order to explore the relationships between these two diseases. Over the last few decades, a large number of genetic data have been generated using next generation sequencing (NGS) technology and these can be used to decipher the genetic relationships between the two types of diseases. Therefore, we have conducted a meta-analysis of NGS data to explore evidence concerning the genomic and functional relationships of endometriosis and cancers in women.

## MATERIALS AND METHODS

### Endometriosis-related gene curation from the literature

To explore the genetics of endometriosis, we conducted a literature search and extracted 3270 records from the GeneRIF (Gene Reference into Function), a database providing lists of genes and their functions (*Maglott et al., 2010*). We curated the literature data manually to extract the corresponding gene names. In total, we curated 984 genes related to studies of endometriosis and used these as a basis to explore similarities in the genetic mechanisms of cancers.

### Differential gene expression for endometriosis dataset

To obtain independent evidence at the expression level, we focused on those studies which compared endometriosis patients with healthy individuals We downloaded two datasets (GSE2339 and GSE5108) related to endometriosis that were collected from the gene expression omnibus (GEO) database, a repository of high-throughput gene expression data of original submitter-supplied records and curated records. Using GEO2R, we compared endometriosis samples with that of healthy individuals and, using the data filtering criteria of adjusted $P$ values <0.05 and an absolute fold change >2, identified genes that are differentially expressed (*Barrett et al., 2012*). Overlapping analysis was conducted to identify genes common to the two GEO datasets and the curated endometriosis gene lists.

## Mutational analysis

The cancer related mutational analyses were conducted using the cBio portal (*Cerami et al., 2012*) which enables the exploration, visualization and analysis of multidimensional cancer genomics data. We selected the TCGA datasets (*Tomczak, Czerwińska & Wiznerowicz, 2015*) of cancers in women including ovarian serous cystadenocarcinoma, cervical squamous cell carcinoma, breast invasive carcinoma, and endometrial carcinoma in order to analyse the mutational frequency of the genes that are related to endometriosis. The combined studies dataset contained a total of 7,462 samples.

## Functional enrichment and network analysis

To investigate the biological systems in the endometriosis-related genes, we analysed the genes common to endometriosis and the differential expression of endometriosis by using the online tools Toppfun (*Chen et al., 2009*), REVIGO (*Supek et al., 2011*) and GeneMania (*Warde-Farley et al., 2010*). The molecular functions of the key endometriosis-related genes were analysed using ToppFun. ToppFun is a web tool which allows users to explore the molecular functions of gene ontology (GO) including cellular components, biological processes and molecular functions. We extracted gene IDs and their corresponding *P* values for the visualization process using REVIGO. REVIGO summarized and removed the redundant GO terms from a long list. The GO results served as input data in REVIGO and provided a semantic similarity-based scatterplot of GO terms from Toppfun. To perform the network analysis we used GeneMania to identify the interactions of the selected genes and Cytoscape (*Shannon et al., 2003*) to characterize and visualise the network results.

# RESULTS

## Identification of potential endometriosis-related genes

In order to explore the association between endometriosis and four cancers of women, we identified the genes commonly implicated in those diseases (Fig. 1). To achieve this, we manually curated endometriosis-related genes from 3273 PubMed abstracts (Table S1). The curated results were mapped based on their official gene names in concordance with data integration for other cancer genomics data sources. In total, we curated 984 endometriosis genes. To further refine the key genetic factors related to the endometriosis, we performed differential expression analyses on two endometriosis-related expression datasets: GSE2339 and GSE5108. GSE2339 is a two disease (endometrioma and eutopic endometrium ) state set presenting the molecular mechanisms underlying the pathology of endometriosis (*Hawkins et al., 2011*). GSE5108 presents the gene expression profile of endometriosis (*Eyster et al., 2007*).

Using a cut-off as the adjusted *P* value <0.05 and the absolute fold change >2, we identified 1,037 genes associated with endometriosis (Table S2) from GSE2339 and 800 from the GSE5108 datasets. The overlapping results revealed that 165 (39+65+61) genes (Fig. 1A) shared at least one GEO dataset and curated gene list and, of these, 65 and 61 genes are common to GSE2339 and GSE5108, and 39 are in both GEO datasets. Using these lists, we explored the key molecular processes for the endometriosis. In summary,

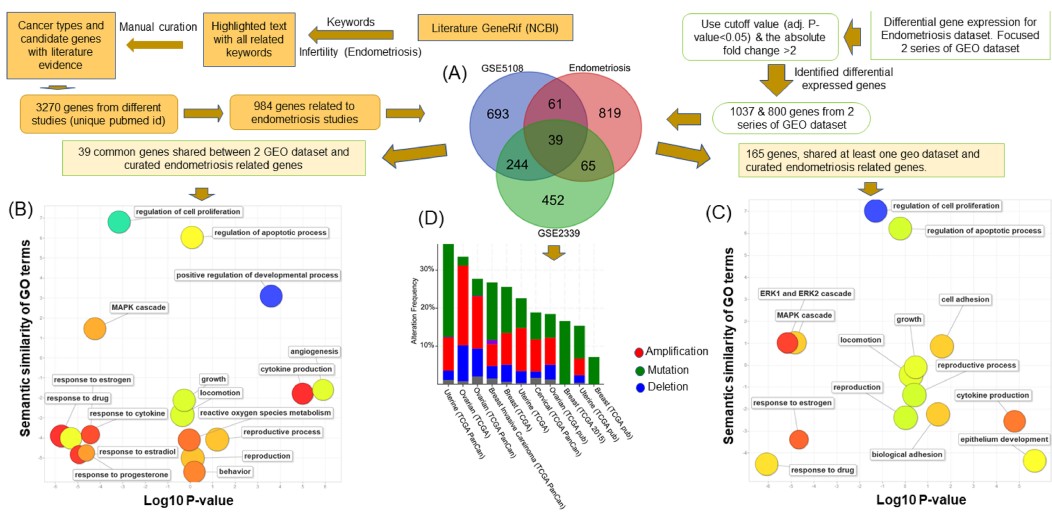

**Figure 1 Pipeline to identify the association between endometriosis disease and four cancers of women.** The flow chart shows the pipeline to discover the common implicated genes in endometriosis disease and four cancers of women. This process had several steps. The first (A) involved identification of endometriosis-related genes from GeneRif and through manually curating data from the literature. This enabled us to extract the corresponding gene names in human and overlapping analysis of curated endometriosis-related genes and two sets (GSE2339 and GSE5108) of the GEO dataset. Thirty-nine genes were common to the curated endometriosis-related gene and two datasets of GEO and 165 genes common to endometriosis-related gene and one set of GEO. (B) Functional enrichment analysis of 39 genes common to three data sets. (C) Functional enrichment analysis of 165 (39 + 65 + 61) genes shared by at least one GEO dataset and curated endometriosis gene list. The scatterplots of (B) and (C) present the summarized GO terms of all endometriosis-related genes. Circles show the GO clusters and are plotted in two-dimensional space according to other GO term semantic similarities. $Y$-axis demonstrates the similarity of the GO terms; $x$-axis indicates the log of corrected $P$-value; circle colour represents the frequency of the GO term in the Gene Ontology Annotation (GOA) database. (D) The cancer type summary of the 39 genes. The $Y$-axis shows the alteration frequency in percentage (including both amplification and deletion mutation); the $x$-axis indicates the cancer types. Blue, Deletion; Red, Amplification; Green, Mutation; and Grey, Multiple Alterations.

our computational workflow integrated the literature curated endometriosis genes with two other GEO endometriosis datasets.

## Functional analysis of major endometriosis-related genes common in GEO datasets and curated list

Functional enrichment analysis of the 165 genes shared by our curated endometriosis-related gene list and at least one GEO dataset was conducted using Gene Ontology (GO) terms as functional units (Fig. 1C). The results revealed the genes enriched with cell proliferation (GO:0042127; $P$-value: 1.03E–41), growth (GO:0040007; $P$-value: 2.17E–21), apoptotic process (GO:0006915; $P$-value: 5.54E–21), and cell adhesion (GO:0007155; $P$-value: 1.47E–17) (Table S3). All these gene functions are associated with cancer progression. In addition, we identified hormone stimulating genes involved in the occurrence of endometriosis. Eighteen genes were found to be involved in the responses to estrogen ($P$-value = 2.45E–12), five (*PTGER4, PTN, THBS1, CCL2, C3*) to progesterone ($P$-value =7.45E–05) and 45 to cytokine ($P$-value = 9.567E–24). Thirty-one genes were
associated with cytokine production (*P*-value =7.03E-14) and 17 genes involved in the *ERK1* and *ERK2* cascades (*P*-value =5.27E–10) (Table S3). In addition, biological adhesion (GO:0022610; *P*-value = 2.04E–17) and *MAPK* cascade (GO:0000165; *P*-value = 2.00E–17) play important roles in cell development processes related to cancer progression.

We conducted another functional enrichment analysis using ToppFun to explore the functions of the 39 genes common to both GEO data sets and the curated endometriosis gene list (Fig. 1B). This showed that the genes are enriched in most of the vital processes in cancer and tumor progression including: apoptotic process (GO:0006915; *P*-value =3.47E–06), the regulation of the cell cycle (GO:0051726; *P*-value =4.48E), cell proliferation (GO:0042127; *P*-value = 1.26E–11), cell death (GO:0010941; *P*-value =4.95E–07), and cell differentiation (GO:0045595; *P*-value = 1.06E–07). Our study also found that five genes are also involved in the enrichment of estrogen (GO:0043627; *P*-value = 1.22E–04) and four in the enrichment of progesterone (GO:0032570; *P*-value = 3.53E–06). Ten genes were found to be associated with angiogenesis (GO:0001525; *P*-value =2.79E–08), twelve genes in responses to cytokine (GO:0034097; *P*-value = 6.83E–08) and seven genes in cytokine production (GO:0001816; *P*-value = 5.54E–04). Eleven genes were involved in the *MAPK* pathway (GO:0000165; *P*-value =2.05E–06) and five genes were involved in *ERK1* and *ERK2* cascade (GO:00703; *P*-value =3.00E–04) (Table S4).

As presented in Tables S3 and S4, our study identified that *C3* is common in both estrogen, progesterone, angiogenesis, cytokine production, *MAPK* pathway, *ERK1*, and *ERK2*. Pleiotrophin (*PTN*), a member of a highly conserved human gene family (*Rauvala, 1989*), is common in the apoptotic process, cell cycle, cell proliferation, cell death, cell differentiation, progesterone, estrogen, and angiogenesis. *LEP* is common in the apoptotic process, cell cycle, cell proliferation, cell death, cell differentiation, estrogen, angiogenesis, cytokine, cytokine production, and *MAPK* pathway. Both *CCL2* and *THBS1* are involved in the apoptotic process, cell proliferation, cell death, progesterone, angiogenesis, cytokine production, and *MAPK* pathway. *AGTR1* is involved in the apoptotic process, cell proliferation, cell death, cell differentiation, estrogen, and cytokine production. *ESR1* is involved in cell cycle, cell proliferation, cell death, estrogen, and the MAPK *ERK1* and *ERK2* pathways. *PLCB1* and *CSF1R* are concerned in the cytokine and MAPK pathways. *GATA2, SOX17, DCN, PTGIS*, and *NRP2* are involved in angiogenesis; *AIF1, CXCL16, IRF6, MME, FOXA2, DCN, PTGIS, TNFSF13B* in cytokine; and *NGF, PLA2G2A, LPAR3*, and *NTF3* in the MAPK pathway only.

## Interconnectivity of Endometriosis-related genes

In order to identify the interconnectivity of endometriosis-related genes, we conducted a network analysis of the 165 genes which shared at least one GEO dataset (Fig. S1) and the curated gene list of endometriosis. We performed network analysis, using GeneMania and Cytoscape, and identified correlations among the genes of interest. To avoid the potential of a large number of non-significant correlations, we used only reliable interactions (*Cerami et al., 2010*). Additionally, we compared the results from the 165 genes with another network analysis focusing on 39 common genes of curated endometriosis genes and both GEO data sets.

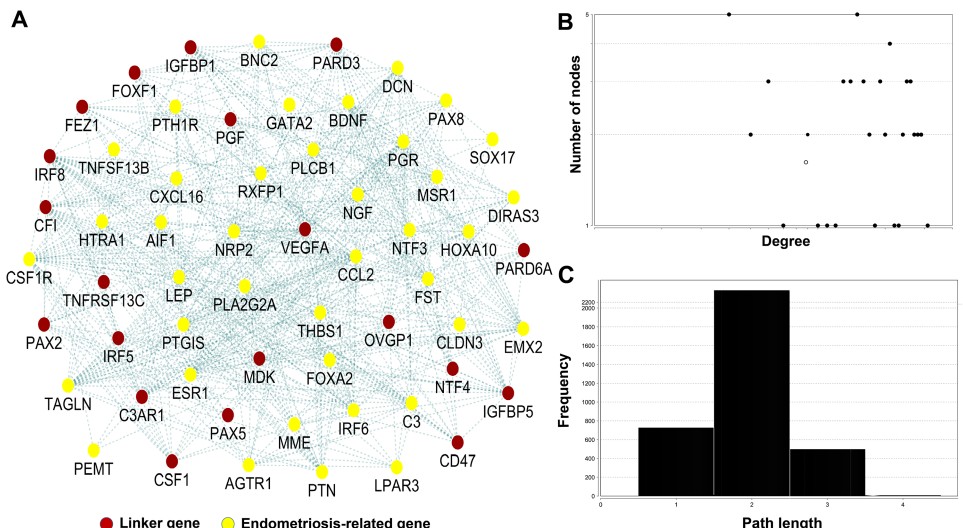

**Figure 2** **Network analysis of 39 genes common in endometriosis-related genes and two sets of GEO dataset.** (A) The network represents the molecular function-based relationship between these 39 genes and the linker genes in endometriosis and cancer development. The red circle indicates the linker gene and yellow circle indicates endometriosis-related gene. (B) The degree distribution. (C) The short path length frequency.

The network analysis of the 165 endometriosis related genes showed that most of the genes has more than 20 connections. We found maximum connections (75) in *SPARC* followed by *FN1* (72), *VCAM1* (61), *DAPK1* (54), *CD14* (53), *HLA-DRA* (53), *TFAP2C* (51), *CCR1* (50), *CDH3* (47), *CXCL12* (46), *TIMP1* (46), *BGN* (45), *CSF1R* (43), *MMP9* (43), *CNN1* (41), *LIPC* (41), *CDH1* (40). From the derived network, we identified novel linker genes showing functional links with the endomitriosis related genes. Among the identified linker genes, twenty genes (*CXCL9, TNFAIP6, FAP, A2M, TIMP3, CCL18, FCER1G, MMP2, CD68, C1QB, TNC, FPR1, CCL5, C3AR1, IGF1, ACTA2, TYROBP, MMP1, AOAH* and *COL3A1*) have more than >20 connections. Results suggested that these linker genes can be used in prognostic studies of endometriosis.

A network analysis of 39 genes is presented in (Fig. 2A). By focusing on genes with the highest number of interactions, we found eight genes with 10 or more connections. Gene *TAGLN* had highest number of connections (23) followed by *CSF1R* (17), *ESR1* (16), *THBS1* (16), *IRF6* (14), *PLA2G2A* (12), *BDNF* (10), and *PLCB1* (10). We also revealed 20 novel linker genes (*VEGFA, NTF4, PAX2, PAX5, CD47, CF1, CSF1, C3AR1, IGFBP1, PGF, FOXF1, OVGP1, IRF8, IRF5, MDK, PARD6A, PARD3, TNFRSF13C, FEZ1*, and *IGFBP5*) connected with the 39 genes. Among the linker genes, 16 genes had 10 or more connections with highest in *CFI* (25) followed by *C3AR1* (24), *IGFBP1* (21), *VEGFA* (19), *IRF8* (18), *IGFBP5* (18), *CSF1* (15), *PARD3* (15), *FEZ1* (14), *FOXF1* (14), *OVGP1* (13), *PARD6A* (13), *CD47* (12), *PGF* (12), *IRF5* (10), and *PAX2* (10).

The network topological analysis identified 59 gene nodes and 364 gene-gene interactions, which showed that the majority of the gene nodes have multiple connections (Fig. 2B). Of the 59 nodes, 39 were identified from our gene list and the remaining 20

were linker genes. The degrees of the nodes in the map fit a power law distribution $y=ax^b$, where $a = 2.684$ and b is an exponent with an estimated value of $-0.110$. The correlation between the given data point and the corresponding point on the fitted curve is 0.195 ($R^2 = 0.014$). Topological analysis on the shortest path length distribution analysis shows that the average length of the shortest path is 2, which indicates that the number of nodes which instantly connected (Fig. 2C).

## Mutation frequency of major endometriosis-related genes common in three data sets

We conducted a mutational analysis of the 39 listed genes of endometriosis using TCGA datasets associated with four cancers of women. For mutational analysis, we used 7,462 TCGA samples collected from 11 studies, consisting of 3,834 breast, 1,754 ovarian, 1,577 endometrial and 297 cervical cancers. The frequency of alteration in cBio Cancer Genomics Portal is defined by mutation, copy number amplification, and homozygous deletion in tumor samples (*Cerami et al., 2012*). Alteration frequency of the TCGA samples revealed that all the 39 genes have a high alteration rate in the tumor samples as revealed by gene amplifications. For example, of the 585 cases of TCGA ovarian epithelial tumor, the frequency of genetic alterations was 60.17% of which the highest alteration was observed in amplification (37.44%, 219 cases) followed by deep deletion (10.6%, 62 cases) (Fig. 3C). In 1,169 cases of ovarian cancer, we observed 54.06% alteration and 41.75% of these were due to amplification. For endometrial carcinoma, the frequency was tested in 529 cases and 59.74% showed genetic alteration with maximum of these being in mutation (34.85%). For invasive breast carcinoma TCGA cohort (of 1086 cases), 40.88% of all were genetically altered and 21.73% of these were due to amplification. The cervical squamous cell carcinoma had 41.83% genetic alteration from 251 cases with highest number due to amplification (17.93%) (Fig. 3E). These results showed the importance of endometriosis-related genes in the development of cancers in women as a consequence of their function in promoting a large number of copy number gains (*Wee et al., 2018*).

In addition to the sample analysis, we explored the genomic alterations in multiple genes across several tumor samples (Figs. 3A–3D). We used the OncoPrint in cBioportal for a query search of alterations in the 39 endometriosis-related genes in TCGA ovarian serous cystadenocarcinoma, TCGA breast invasive carcinoma, TCGA uterine corpus endometrial carcinoma, and TCGA cervical squamous cell carcinoma tumor samples. An oncoprint is a graphical display of gene mutations in human cancer tumor samples. We further identified the top 10 genes with the highest amplification rate in samples of four cancer types. (Figs. 3A–3D). For ovarian serous cystadenocarcinoma, there was a total of 12 (*MME, AGTR1, SOX17, PTN, NTF3, MSR1, PGR, PLCB1, BNC2, PTGIS, LEP,* and *PAX8*) genes with >4% of genetic alterations frequency. *IRF6* showed 6% and *SOX17, MSR1, PTGIS* showed 3% in the TCGA breast invasive carcinoma tumor samples. In TCGA uterine corpus endometrial carcinoma samples, there were nine (*SOX17, C3, PAX8, ESR1, MME, MSR1, BNC2, IRF6,* and *PGR*) genes with >5% alteration frequency. Three genes (*MME, PGR,* and *AGTR1*) showed >7% alteration frequency in the TCGA cervical squamous cell carcinoma. Most of these alterations were related to amplification. *MME, SOX17, AGTR1, PGR,* and *ESR1*

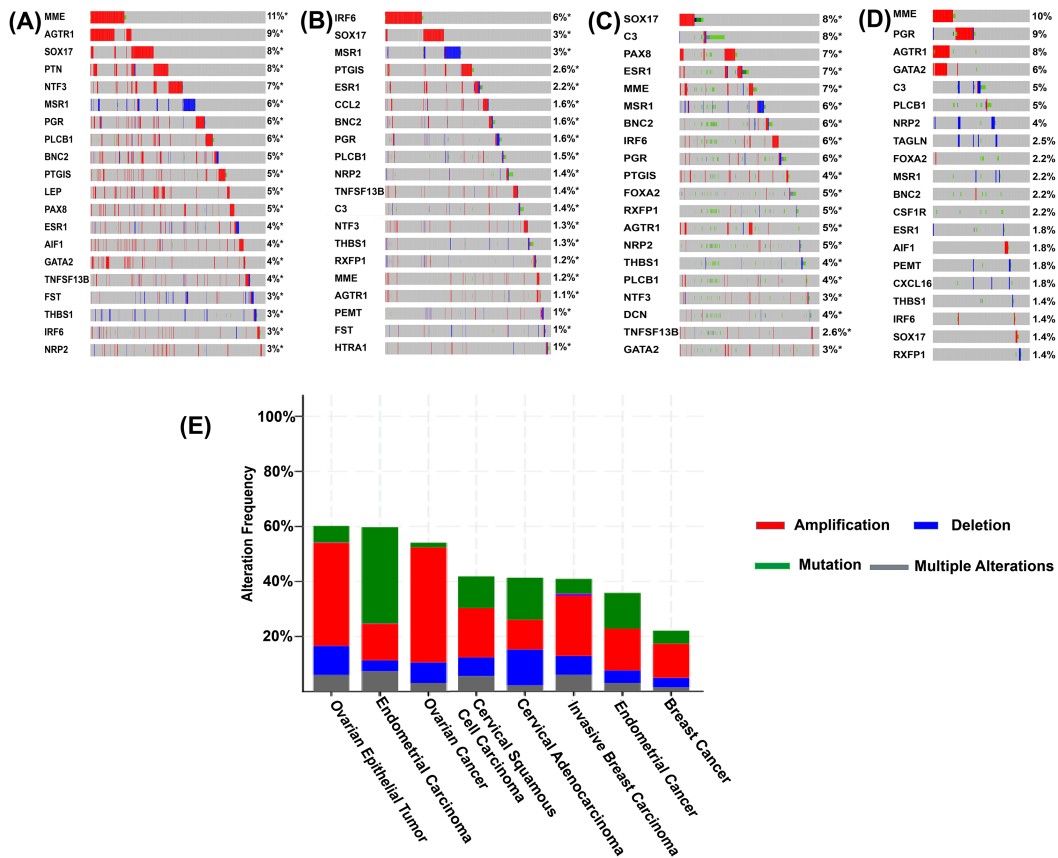

**Figure 3** **Sample-based mutational patterns for the twenty genes with the highest amplification rate.**
(A) TCGA ovarian serous cystadenocarcinoma; (B) TCGA invasive breast carcinoma; (C) TCGA endometrial carcinoma; (D) TCGA cervical squamous cell carcinoma. Columns indicate samples and rows indicate genes. The colour bars are used to represent the genomic alterations. (E) The cancer type summary for the 39 common genes across TCGA women-specific cancers tumour samples as shown by alteration frequency based on amplification. $Y$-axis shows the alteration frequency in percentage (including both amplification and deletion mutation); $x$-axis indicates the cancer types. Blue, Deletion; Red, Amplification; Green, Mutation; and Grey, Multiple Alterations.

had the highest amplifications frequency ranging from 4% to 11% in ovarian serous cystadenocarcinoma. We observed similar features in the uterine corpus endometrial carcinoma dataset: the genes with the highest frequency included *SOX17* and *C3* (8%), *PAX8, MME* and *ESR1* (7%). In breast invasive carcinoma, *IRF6* showed the highest rate of alteration (6%) followed by *SOX17* and *MSR1* (3%). In the case of cervical squamous cell carcinoma, alteration frequency was greatest in *MME* (10%), followed by *PGR* (9%) and *AGTR1* (8%). Therefore, all these endometriosis-related genes are associated with gene amplification events across the four cancers of women TCGA samples.

## Mutational analysis of linker genes involved in the endometriosis network

We applied the network analysis and constructed a gene network to identify the global connections of the 39 genes identified by overlapping analysis between endometriosis-related genes and two GEO datasets. The derived network comprised 20 linker genes that were shown to connect with core genes. Interestingly, one of those linker genes (*C3AR1*) is within the 165 full gene list, which was identified by overlapping analysis between literature-based endometriosis-related genes and one set of GEO datasets.

The additional mutational analysis on the 20 linker genes, including *C3AR1* (Fig. 4A), showed a significant amplification frequency across tumor samples. Eighty percent of the genes showed more than 1% genetic alterations in eleven TCGA datasets from four cancer types: breast invasive carcinoma, cervical squamous cell carcinoma, ovarian serous cystadenocarcinoma, and uterine corpus endometrial carcinoma. The TCGA ovarian serous cystadenocarcinoma patients had >40% genetic alteration/expression (median) (Fig. 4B). Overall, most of cancer cohort patients showed >20% genetic alterations. Based on 7462 samples, the greatest rate of alteration was observed in *PARD3* and *C3AR1* (both 3%), followed by *CD47* (2.6%) and *IRF8* (2.4%). The graphical presentation of expression result from cBioportal (Fig. 4B) revealed that the maximum expression of *C3AR1* was found in breast cancer samples followed by ovarian then cervical cancers and minimum in uterine or endometrial cancers.

## Mutational and functional analysis of common genes involved in endometriosis and cancers in women

Overlapping analysis of 165 endometriosis genes and previously identified 52 genes of four women's cancers (*Bhyan et al., 2019*) revealed that nine genes were common: *SPARC, CDH1, MET, TIMP1, BRCA1, IGF2, PGR, MMP9* and *ESR1* (Fig. S2). *PGR* and *ESR1* are also found in the list of the common 39 genes from the three endometriosis-related data sets. Further analysis revealed that all the nine genes are frequently mutated and the frequency varied from 1% to 8% (Fig. 5A). Gene *CDH1* had the highest rates of alteration (8%) followed by *BRCA1* (4%), *PGR* (4%) and *ESR1* (4%). *IGF2* had the lowest rate of alteration (1%). TCGA data sets collected from the eleven studies provided 7,462 samples in which the maximum rate of genetic alteration occurred in endometrial carcinoma (36.86% of 529 cases), followed by ovarian epithelial tumor (27.69% of 585 cases), invasive breast carcinoma (26.7% of 1,086 cases), and ovarian cancer (26.26% of 1,169 cases) (Fig. 5B).

Functional enrichment analysis revealed that the nine genes were associated with a number of key cancer pathways and reproductive system biological processes including: the progesterone receptor signalling pathway (GO:0050847; *P*-value =2.90E–03); the hormone-mediated signalling pathway (GO:0009755; *P*-value = 1.46E–04); and the intracellular receptor signalling pathway (GO:0030522; *P*-value = 3.20E–04) (Table S3). Among the nine genes, *PGR* and *ESR1* were involved in all three pathways and *PGR* alone was involved in the progesterone receptor signalling pathway. These results suggested that endometriosis and four women's cancers arise from common genetic mutations.

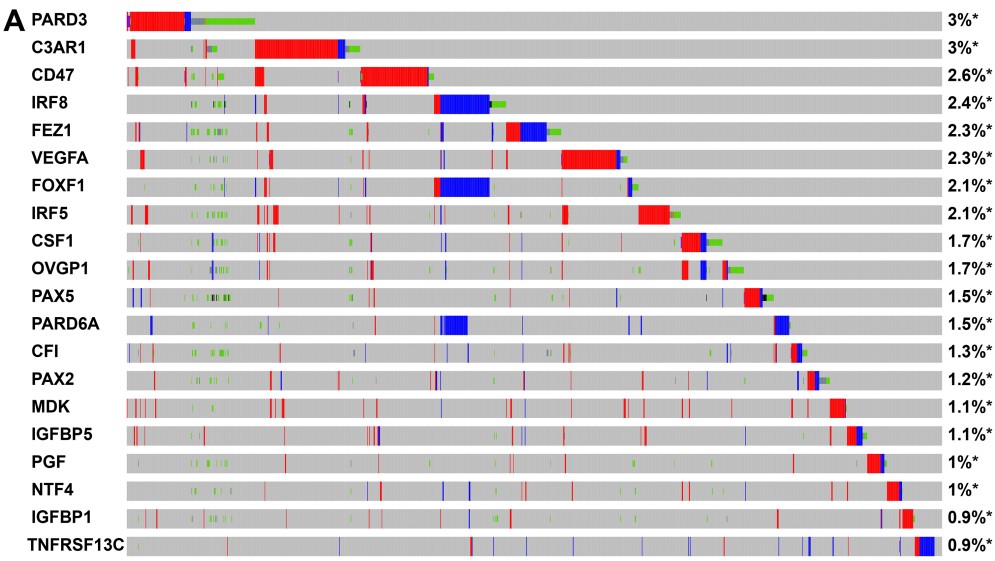

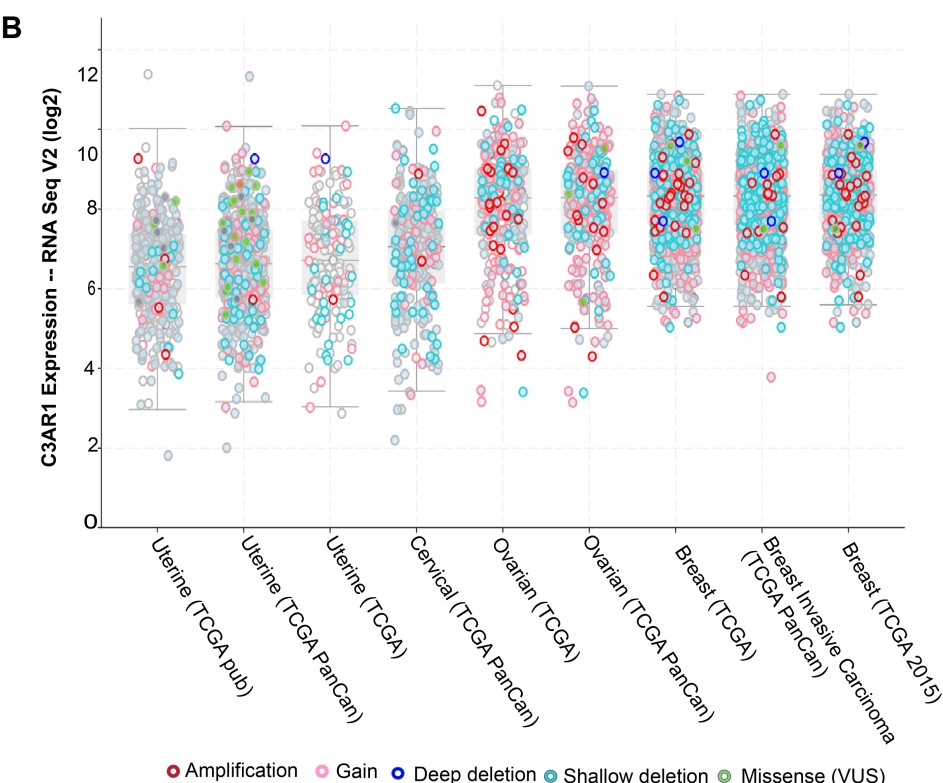

**Figure 4** **The mutational landscape for the twenty linker genes in TCGA tumour samples of different cancers by up-regulated gene expression.** The coloured bars represents the genomic alterations. (A) The different mutational types are marked using different colours. (B) The expression of C3AR1 based on RNA seq V2 profile of cBioprtal oncoprint result in TCGA ovarian cystadenocarcinoma, invasive breast carcinoma, endometrial carcinoma, and cervical squamous cell carcinoma. The analysis sorted by a median that presented expression profile with a log scale, mutation and copy number alterations. The red and blue bars show the amplification and deletion, respectively. The grey bar indicates no mutations in the sample.

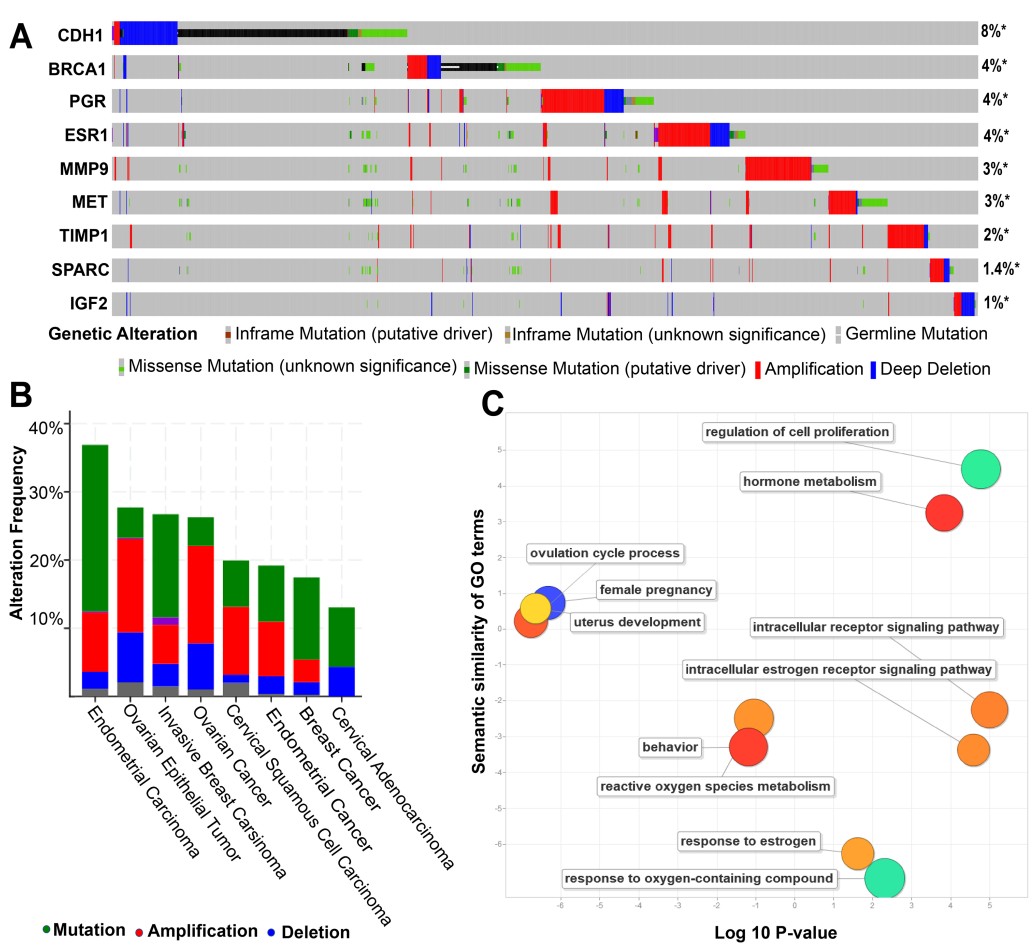

**Figure 5  Sample-based mutational and functional analysis for the nine genes common to endometriosis and four cancers of women with high amplification rate.** (A) The sample-based mutational pattern for the nine genes from the four different cancer samples: TCGA ovarian, breast, endometrial and cervical cancers. Columns indicate samples and rows indicate genes. The coloured bar is used to represent the genomic alterations and the percentage represents the alteration frequency for each gene. The different mutational types are marked using different colours: amplification (red) and deletion (blue). (B) A general cancer type landscape between the correlations of genetic alterations based on nine genes. Blue represents deletions; Red indicates amplification. (C) Gene enrichment analysis of nine genes. The scatterplot presents the summarized gene ontology (GO) terms of all nine genes. Circles indicate the GO clusters and are plotted in two-dimensional space according to other GO terms sematic similarities. Circle size shows the log10 *P* value (the larger corrected *P*-value, the smaller circle) and circle colour represents directly proportional to the frequency of the GO term in the GOA database.

## DISCUSSION

Endometriosis is the hormone-dependent abnormal growth of endometrial epithelial and stromal cells outside the uterine cavity which may cause chronic pelvic pain, subfertility and result in an increased risk of ovarian cancer (*Vercellini et al., 2014*; *Saavalainen et al., 2018*; *Pearce et al., 2012*). In a previous study, we identified some key driver genes of four cancers of women which function as hormonal stimulants (*Bhyan et al., 2019*). But, the high-grade serous ovarian cancers data set from TCGA are not strongly associated with endometriosis

and the clear cell and endometrioid OCs that are associated with OC are not represented in TCGA. However, *Dawson et al. (2018)* suggested that endometriosis and four cancers of women may share common biological mechanisms. Previous studies demonstrated that there was an increased risk of ovarian and breast cancer due to endometriosis (*Saavalainen et al., 2018*; *Pearce et al., 2012*; *Saavalainen et al., 2019*). Our research involved a meta-genomic study on endometriosis-related genes and an assessment of their involvement in four cancers in women: breast, cervical, ovarian and endometrial. Integrated analyses, including functional enrichment, network, and mutational analysis provided a list of key genes playing dual role in endometriosis and women's cancers.

In this study, functional analysis of endometriosis-related genes revealed shared mechanisms of endometriosis concerned with four cancers of women. This was particularly the case for cell proliferation, growth, apoptotic process, cell adhesion, regulation of cell cycle, regulation of cell death, and regulation of cell differentiation. The control of cell proliferation and apoptosis are the key regulatory mechanisms of cancer progression. The apoptotic process is highly regulated in human cells and essential for maintaining the physiological balance between cell death and cell growth (*Koff, Ramachandiran & Bernal-Mizrachi, 2015*). Different extracellular and intracellular signals, including hormones, growth factors and cytokines, can stimulate the different pathways (*Dhillon et al., 2007*) and regulate cell differentiation, growth and apopotosis (*Kim & Choi, 2010*). We identified common hormone stimulant pathways, including enrichment of estrogen and progesterone, angiogenesis, response to cytokine, and cytokine production, as functional roles of endometriosis-related genes. Estrogen and progesterone are the key hormones involved in reproductive development but are also associated with tumor growth and the spread of some cancers (*Subramani et al., 2017*). It has been shown that estrogen exposure is directly associated with an increase in the risk of developing breast cancer (*Begg et al., 1987*; *Pike et al., 1979*), whereas reducing exposure is thought to be protective against breast cancer (*Hulka, 1997*). Estrogen also increases the risk of ovarian cancer, particularly after menopause (*Ho, 2003*). Progesterone and progesterone receptors (PR) are important because of their role as critical regulators of breast and gynecological cancers. Although the uptake of both estrogen and progesterone reduces the risk of ovarian cancer, the mechanisms explaining the role of these two hormones in carcinogenesis is unclear (*Ho, 2003*). Angiogenesis, which is responsible for metastasis in ovarian cancer (*Gavalas et al., 2013*), also has a key functional role in endometriosis. Several studies have indicated that the further outgrowth of ectopic endometrial implants through endometriosis leads to tumor formation (*Shubik, 1982*). Furthermore, the cancer-modifier cytokines were found to be involved in endometriosis (*Brower, 2005*) and the cytokinins are both tumor necrosis factors (*Esquivel-Velázquez et al., 2015*) as well as being associated with a number of gynaecological cancers (*Heikkilä, Ebrahim & Lawlor, 2008*; *Murooka, Ward & Fish, 2005*).

Our study showed that kinase signalling pathways, such as *MAPK* (mitogen-activated protein kinases), *ERK1* and *ERK2*, are activated in endometriosis. *MAPK* act as integration points for many biochemical signals and are involved in a variety of cellular processes, including cell proliferation, differentiation, transcription regulation and development (*Imajo, Tsuchiya & Nishida, 2006*). In association with several environmental stimuli such

as, hormones and cytokines play a role in activating *MAPK* pathways (*Imajo, Tsuchiya & Nishida, 2006*). The dysregulation of protein kinase stimulated by several oncogenic driver mutations was found to accelerate uncontrolled cellular proliferation in kinase-dependent tumour growth (*Burotto et al., 2014*; *Sawyers, 2003*). *ERK1* and *ERK2* are two extracellular regulated kinases and the final effectors of the *MAPK* pathway (*Robinson & Cobb, 1997*; *Liu et al., 2018*). In addition, they are both regulators of malignant breast cancer cells (*Milde-Langosch et al., 2005*). Therefore, these kinases were identified as potential targets for the treatment of cancers and endometriosis. Endometriosis-related genes identified in this study were found to be associated with key biological mechanisms controlling cancers in women. *C3* is a plasma protein which increases cell proliferation once synthesized in malignant ovarian epithelial cells (*Cho et al., 2014*). *PTN* is a member of a highly conserved human gene family (*Rauvala, 1989*) and is a key gene in the process of endometriosis and regulates multiple functions, including apoptosis, cell cycle, cell proliferation, cell differentiation, progesterone and estrogen production, and angiogenesis (Table S4). Overexpression of *PTN* in breast cancer cells, such as *MCF-7* gene enhanced angiogenesis in the rabbit corneal assay (*Choudhuri et al., 1997*). In addition, a truncated form of *PTN* was shown to act as a dominant-negative effector on the proliferation and angiogenesis of breast cancer cells, *in vitro* and *in vivo* (*Ducès et al., 2008*). Another endometriosis-related gene, *LEP*, functions as a key mediator in obesity-associated cancers including breast, colorectal and prostrate (*Renehan, Soerjomataram & Leitzmann, 2010*).

From the network analysis of 165 and 39 genes, we identified only one common linker gene, *C3AR1*, although there is no evidence of its involvement with endometriosis. *C3AR1* is an oncogenic gene that down regulates in tumour cells (*Nabizadeh et al., 2016*; *Yamada et al., 2017*). Formerly, *C3AR1* was considered to be involved in the innate immune response but is now regarded as a factor in cancer (*Opstal-Van Winden et al., 2012*). Additionally, *C3AR1* was found to activate the *PI3K-AKT* pathways that result in cell proliferation (*Cho et al., 2016*; *Towner et al., 2016*). Further investigation of the involvement of this gene in endometriosis and four cancers of women is important for future disease management.

We identified that the alteration frequency was highest (5%) in *SOX17*, which is involved in oncogenesis through tumour suppression, down-regulating *MAML3* expression, modulating nuclear $\beta$-catenin and antagonizing Wnt signaling (*Zhang et al., 2016*). Gene *IRF6*, a transcriptional activator which plays critical roles in endometrial gene expression and the growth and differentiation conceptus trophectoderm (*Fleming et al., 2009*), also showed 5% alteration frequency in our study. High rate (5%) of mutation frequency was observed in macrophage metalloelastase (*MME*), which is a zinc-dependent endoprotease and also involved in *ECM* re-modulation and conversion of plasminogen to angiostatin (*Lavilla-Alonso et al., 2012*). Finally, we discovered eight genes (*MME*, *SOX17*, *AGTR1*, *PGR*, *ESR1*, *PAX8*, *C3* and *IRF6*) with high amplification frequency compared to other genes across the cancer types. These genes are all involved in endometriosis suggesting their involvement in the progression of cancers in women.

## CONCLUSION

Using integrated bioinformatic analysis we discovered evidence of genetic link between endometriosis and women's cancers. We utilized information from next generation sequence data and compiled a list of a large number of endometriosis related genes. Functional analysis confirmed that 39 genes were associated with the processes of tumour formation and cancer progression of which two (*PGR* and *ESR1*) were common to four cancers of women. Mutational analysis proved that eight endometriosis genes had a higher rate of alterations across the four cancers in women. Finally, we explored a novel linker gene, *C3AR1*, which had not been implicated previously in endometriosis. The evidence of shared genetic mechanisms of endometriosis and cancers in women may be an avenue of future disease management and treatment through early diagnosis. This article provides a catalogue of genetic links between endometriosis and cancer as a guide for further investigation and analysis.

## ACKNOWLEDGEMENTS

The authors thank to two anonymous reviewers and the academic editor (Kate Lawrenson) of PeerJ for reviewing the manuscript and providing suggestions to improve the manuscript. We thank Professor Richard Burns (University of the Sunshine Coast) and Dr Mobashwer Alam (The University of Queensland) for their assistance in language editing and putting the manuscript in order.

### Funding

This work was supported by the research start-up fellowship of university of sunshine coast to Min Zhao. The work was supported by the National Natural Science Foundation of China (No. 31871339, 31801120), Guangdong High School Young Innovative Talents Project (No. 2015KQNCX136), Guangdong Science Research Program General Project (No. 20170710123) and Guangzhou Municipal Scientific Research Project (No. 1201630073). The funders had no role in study design, data collection and analysis, decision to publish, or preparation of the manuscript.

### Grant Disclosures

The following grant information was disclosed by the authors:
National Natural Science Foundation of China: 31871339, 31801120.
Guangdong High School Young Innovative Talents Project: 2015KQNCX136.
Guangdong Science Research Program General Project: 20170710123.
Guangzhou Municipal Scientific Research Project:  1201630073.

### Competing Interests

Min Zhao is an Academic Editor for PeerJ.

## Author Contributions

- Salma Begum Bhyan conceived and designed the experiments, performed the experiments, analyzed the data, contributed reagents/materials/analysis tools, prepared figures and/or tables, authored or reviewed drafts of the paper, approved the final draft.
- Li Zhao, Yining Liu and Min Zhao conceived and designed the experiments, authored or reviewed drafts of the paper, approved the final draft.
- YongKiat Wee analyzed the data, authored or reviewed drafts of the paper, approved the final draft.

## Data Availability

Raw data is available in Table S1.

## Supplemental Information

Supplemental information for this article can be found online at http://dx.doi.org/10.7717/peerj.8135#supplemental-information.

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
