# Peer review of "Genetic links between endometriosis and cancers in women"

_PeerJ, doi:10.7717/peerj.8135_

## Round 0.1 · original submission · Minor Revisions

This is a comprehensive study and will be of interest to the endometriosis community. In addition to the suggestions by the reviews, the article needs to be edited throughout for spelling, punctuation and grammar. Also please verify that citations are used appropriately throughout.

The following edits are suggested by the editor:

1. The associations between endometriosis and clear cell ovarian cancer have been overlooked, in the introduction in particular and throughout the manuscript. The associations with clear cell ovarian cancer are in fact stronger than for endometrioid and other subtypes (see Pearce 2012 and Saavalainen 2018; the latter paper is not currently cited in the description of OC associations with endometriosis, but should be). One caveat that should be mentioned in the discussion is that the high-grade serous ovarian cancers the the data set from TCGA are not strongly associated with endometriosis, and the clear cell and endometrioid OCs that are associated with OC are not represented in TCGA.
2. The statement that endometriosis and breast cancer have similar risk factors needs to explained more thoroughly and supported with evidence. Obesity is not a risk factor for endometriosis (in fact, lower BMI has an association with increased risk, although higher BMI may be associated with more severe disease - see doi.org/10.1177/2284026518773939 and others) so this sentence of the introduction needs to be edited. PMID 30883685 is also a relevant citation for the discussion with BC associations.
3. Sentence "Endometrial cancer-associated genes, such as PTEN" - PTEN and the PI3K pathways are also altered in endometriosis-associated ovarian cancer, so this sentence should be removed. Also please

Reviewer 1 ·

Basic reporting

The article by Bhyan et al. deals with the analysis of genetic links between endometriosis and cancers in women. This study represents a very well designed multidisciplinary investigation in this field.
A well-designed functional analysis of major endometriosis-related genes was conducted in an attempt to gain insights for the function of the identified genes as well as the relevant biochemical pathways and processes involved in cancer and tumor progression. Interconnectivity of endometriosis-related genes is another major issue of the article under review, considering that a number of the findings, related to the derived gene networks, may be used in prognostic studies of endometriosis. The authors also pointed out the importance of the high frequency of genetic alterations observed in various endometriosis-related genes. Moreover, another key-issue of this article refers to the functional enrichment performed, which revealed common genetic mutations to be involved in the development of both endometriosis and four types of women cancer.
Overall this study has a very good potential, the results are clear and impressive and, possibly, of high clinical significance at the stage of prognosis. It is well organized, the questions are clear and the approaches followed are straightforward. The entire information is well presented and convincing and the reasoning is logical. The introduction is clear enough with sufficient data and references to support the study hypothesis. The methodology used is correct and appropriate and the quality of writing is sufficient. The tables and figures are very comprehensive and help in summarizing the data and to a better understanding of the findings. Most of the relevant references have been included and detailed information is presented dealing with the current literature.

Experimental design

The methodology used is correct and appropriate and the quality of writing is sufficient.The tables and figures are very comprehensive and help in summarizing the data and to a better understanding of the findings

Validity of the findings

Overall this study has a very good potential, the results are clear and impressive and, possibly, of high clinical significance at the stage of prognosis. It is well organized, the questions are clear and the approaches followed are straightforward. The entire information is well presented and convincing and the reasoning is logical. The introduction is clear enough with sufficient data and references to support the study hypothesis.

Reviewer 2 ·

Basic reporting

English could be improved somewhat as there are several typos. Apart from this, the reporting is fine (although I would suggest citing the recently published meta-analysis by Gandini et al. on cancer risk among endometriosis patients in the Introduction).

Experimental design

The design is sound and methods are described thoroughly, although I fear that many readers could be unfamiliar with the database, software, etc. that were exploited by the Authors. Thus, some more details on this would be helpful.

Validity of the findings

This paper has no very strong conclusions, rather it is kind of an overview of genetic alterations in endometriosis and women's cancers that may serve to generate hypotheses for further study. Because of the nature of this paper, some parts of the Discussion (in particular, the second and third paragraphs) are quite generic and should be tweaked to be rendered more specific to endometriosis.

Additional comments

In general, the authors should be commended for embarking on this review. As stated before, conclusions are not surprising overall, which is somehow to be expected when the main finding is a long list of genes. However, this paper may serve in the future as a sort of catalog of genetic links between endometriosis and cancer, to generate more specific and clear-cut hypotheses to be investigated in dedicated, analytical studies.

---

## Round 0.2 · Minor Revisions

Thank you for revising the manuscript, the paper is much improved as a result. A few minor corrections are requested before the manuscript is suitable for publication:

1. non-requested edit "endometriosis is a common
inflammatory disease" - suggest removal of word inflammatory (which is used more accurately later in the sentence). Although inflammation is a key component, endometriosis is not currently defined as an immune system disorder (such as allergies, IBD, for example)

2. Lines 104-106. Please re-read the Pearce et al manuscript. This paper about ovarian cancer is mis-cited, and included in the description of endometrial (uterine) cancer, when it should be included in the first paragraph. (around line 59). Currently the text reads " A significantly positive association was observed with different histological subtypes of ovarian cancers, particularly with endometroid, epithelial and clear cell types " suggest revise for accuracy to " A significant positive association was observed with different histological subtypes of epithelial ovarian cancers, particularly with endometrioid, and clear cell types ". Please note correct spelling of endometrioid (a subtype of epithelial ovarian cancer).

3. Please correct typos
- line 433-4: "Mutational analysis clearly proved that eight endometriosis genes higher rate OF ALTERATIONS across the four women cancers samples"
- 435 - evidences - should be evidence

---

## Round 0.3 · Minor Revisions

A solid report in an important research area.

I still recommend the section about the associations between ovarian cancer and endometriosis be revised for accuracy before publication. The well proven link between endometriosis and ovarian cancers of clear cell and endometrioid histologies does not come across clearly in its current form.

I maintain the recommendation that spelling, punctuation and grammar need to to undergo correction. Revision by a native English is recommended

e.g. in the abstract alone:
- 'integrating the literature' - recommend remove 'the'
- 'At first' - recommend remove 'at'
- 'DEG evidences' - should be 'evidence' (but would read better if the word analysis was used instead of evidence)
- remove space between ( and PGR
- gene names should be italicized
- "The shared genetic mechanisms of endometriosis and cancers in women identified in this study provided a new avenue of multiple disease management and treatments through early diagnosis." - switch provided to provide. Suggest stating a "possible new avenue", to be more cautious.

---

## Round 0.4 · accepted · Accept

Dear authors

Thank you for your revisions, the manuscript is greatly improved as a result. I made some minor changes to the section relating to links between endometriosis and ovarian cancer for accuracy, these are in the document attached (I can also send it through as a word document if preferred).